# Anticoagulation for Atrial Fibrillation in Patients with Decompensated Liver Cirrhosis: Bold and Brave?

**DOI:** 10.3390/diagnostics13061160

**Published:** 2023-03-18

**Authors:** Irina Gîrleanu, Anca Trifan, Laura Huiban, Cristina Maria Muzica, Oana Cristina Petrea, Ana-Maria Sîngeap, Camelia Cojocariu, Stefan Chiriac, Tudor Cuciureanu, Remus Stafie, Sebastian Zenovia, Ermina Stratina, Adrian Rotaru, Robert Nastasa, Catalin Sfarti, Irina Iuliana Costache, Carol Stanciu

**Affiliations:** 1Department of Gastroenterology, “Grigore T. Popa” University of Medicine and Pharmacy, 700115 Iasi, Romania; 2Institute of Gastroenterology and Hepatology, “Saint Spiridon” University Hospital, 700111 Iasi, Romania; 3Cardiology Department, “Saint Spiridon” University Hospital, 700115 Iasi, Romania

**Keywords:** atrial fibrillation, liver cirrhosis, risk factors, anticoagulation, bleeding, stroke

## Abstract

Atrial fibrillation is frequently diagnosed in patients with liver cirrhosis, especially in those with non-alcoholic steatohepatitis or alcoholic etiology. Anticoagulant treatment is recommended for thromboembolic protection in patients with atrial fibrillation. Considering the impaired coagulation balance in liver cirrhosis, predisposing patients to bleed or thrombotic events, the anticoagulant treatment is still a matter of debate. Although patients with liver cirrhosis were excluded from the pivotal studies that confirmed the efficacy and safety of the anticoagulant treatment in patients with atrial fibrillation, data from real-life cohorts demonstrated that the anticoagulant treatment in patients with liver cirrhosis could be safe. This review aimed to evaluate the recent data regarding the safety and efficacy of anticoagulant treatment in patients with decompensated liver cirrhosis. Direct oral anticoagulants are safer than warfarin in patients with compensated liver cirrhosis. In Child–Pugh class C liver cirrhosis, direct oral anticoagulants are contraindicated. New bleeding and ischemic risk scores should be developed especially for patients with liver cirrhosis, and biomarkers for bleeding complications should be implemented in clinical practice to personalize this treatment in a very difficult population represented by decompensated liver cirrhosis patients.

## 1. Introduction

There has been an increasing prevalence of atrial fibrillation (AF) over the last few years with a negative impact on morbidity and mortality [1]. AF is one of the most common arrhythmias among patients with liver cirrhosis (LC) [2]. The prevalence of AF in LC patients has been reported to be 11.6%, higher in decompensated LC and patients with alcoholic liver disease or non-alcoholic steatohepatitis (NASH), elderly, or associating comorbidities such as diabetes mellitus, arterial hypertension, chronic obstructive pulmonary disease, and chronic kidney disease [1]. Cirrhotic patients have been shown to have an increased risk of stroke compared with cirrhotic patients without AF (1.1% vs. 1.6%, *p* < 0.0001) [1] in direct relation to LC severity, and higher mortality [3]. A recent meta-analysis demonstrated that, when compaing the same age groups, the prevalence of AF in patients with LC is higher than the prevalence described in the general population (5–7% vs. 2–4%), with a 1.44-fold increase in mortality [3,4].

The current AF guidelines recommend anticoagulant treatment for thromboembolic complication prevention in men with CHA2DS2-VASc scores ≥2, and women with CHA2DS2-VASc scores ≥3. For patients with a moderate thromboembolic risk, the anticoagulant treatment should take into consideration the risk of bleeding complications [5,6,7].

Oral anticoagulants, direct oral anticoagulants (DOACs), and vitamin K antagonists (VKAs) markedly reduced the risk of stroke, and they decreased mortality in patients with AF [8,9,10]. Randomized controlled trials (RCTs) have demonstrated an annual rate of stroke of 1.5% and a mortality rate of 3% in patients with AF and anticoagulant (AC) treatment [8,11]. The main cause of death was related to the severity of cardiac failure or sudden death [12,13] and not related to ischemic complications. Even if patients with LC and AF are associated with an increased risk of thromboembolic complications, they have been excluded from the pivotal studies that have demonstrated the efficacy and safety of AC treatment in AF, considering the higher bleeding risk than the benefits of thrombotic events prevention, especially in patients with decompensated LC [8].

Recently, all the above-mentioned data have been reconsidered, since studies have demonstrated that cirrhotic patients are not “auto-anticoagulated”, moreover they could be predisposed to a hypercoagulable state, the balance between bleeding and thrombosis being very fragile, especially in decompensated LC [14,15,16]. Recent studies have demonstrated that AC treatment in patients with compensated LC is safe and has the same efficacy for stroke prevention as in the general population, although there are less data regarding the safety of AC treatment in patients with decompensated LC [16,17,18]. Taking into account all these data, liver cirrhosis is still considered to be a non-modifiable bleeding risk factor among patients with AF [8], and all DOACs are not indicated in patients with Child–Pugh class C liver cirrhosis because there are few data regarding efficacy and safety in this special population.

This review aims to evaluate the efficacy and safety of anticoagulant treatment in patients with decompensated liver cirrhosis and atrial fibrillation.

To evaluate the risks of thromboembolic complications and bleeding in patients with cirrhosis receiving oral anticoagulation for AF, we performed a systematic literature search on MEDLINE (1990 through January 2023), EMBASE (1990 through January 2023), and the Cochrane Database of Systematic Reviews. Three investigators (I.G., A.R., and T.C.) independently conducted the systematic literature review using a search strategy that included the terms “atrial fibrillation”, “anticoagulation”, and “liver cirrhosis.” Only observational studies (cohort, case-control, or cross-sectional studies) reporting the outcomes of interest (thromboembolic complications and bleeding) in cirrhotic patients using oral anticoagulation for AF qualified as eligible research.

## 2. Physiopathology and Risk Factors of Atrial Fibrillation in Liver Cirrhosis

### 2.1. Physiopathology of Atrial Fibrillation in Liver Cirrhosis

Liver cirrhosis is the end stage of all chronic liver diseases and it is characterized by parenchymal destruction due to liver fibrosis and chronic inflammation. The most frequent causes of cirrhosis in industrialized countries are chronic viral hepatitis and heavy alcohol use. Obesity, linked to non-alcoholic steatohepatitis, is increasingly recognized as a primary or contributing factor in chronic liver disease leading to cirrhosis, whether alone or in conjunction with alcohol, hepatitis C, or both. Primary biliary cholangitis, metabolic disorders, and autoimmune hepatitis are some additional causes of cirrhosis. Liver cirrhosis is characterized by a complex hemodynamic and inflammatory dysfunction in direct relation to liver fibrosis and portal hypertension (Figure 1). Patients with LC are characterized by sympathetic and parasympathetic dysfunction associated with heart rate variability and rest tachycardia and prolonged QT interval [19]. Cirrhotic patients also have systemic vasodilatation due to increased levels of nitric oxide (NO), in contrast with intrahepatic vasoconstriction secondary to increased endothelin levels and decreased NO levels. The consequences of these hemodynamic abnormalities are activation of the rennin–angiotensin–aldosterone system and development of hyperdynamic circulation and cirrhotic cardiomyopathy [19]. Moreover, portal hypertension leads to an increase in intestinal permeability and a high level of pathogen-associated molecular patterns, stimulating the secretion of proinflammatory cytokines such as tumor necrosis factor-α, interleukin-8, interleukin-6, and fibrogenic factors such as galectin-3, the common element between liver and heart fibrosis [20,21,22,23,24]. Vasoactive neuropeptides, such as vasoactive intestinal peptide (VIP), have elevated levels in decompensated LC as a result of a decreased hepatic turnover [25]. VIP is implicated in the onset of AF thru the vagal nerve.

Abbreviations: IL-1, interleukin-1; IL-6, interleukin-6, IL-1β: interleukin-1β; PAMPs, pathogen-associated molecular patterns; TNF-α, tumoral necrosis factor.

All the hemodynamic changes associated with portal hypertension also reflect on cardiac morphology, and cirrhotic patients have an enlarged left atrium and diastolic dysfunction [26].

The physiopathological abnormalities associated with liver cirrhosis are upregulated by the degree of portal hypertension, and they are accentuated by the severity of liver cirrhosis, explaining the higher prevalence of AF in patients with LC in correlation with the model for end-stage liver disease (MELD) score [22].

### 2.2. Risk Factors for Atrial Fibrillation in Liver Cirrhosis

All the factors demonstrated to be potentially involved in AF in cirrhotic patients are known to predispose to AF in the general population, although two main risk factors should be mentioned in particular in these patients: non-alcoholic steatohepatitis etiology of LC and chronic alcohol consumption.

In NASH liver cirrhosis, there are many cardiovascular risk factors associated with an increased risk of AF. Atherogenic dyslipidemia, hyperhomocysteinemia, central obesity despite overall sarcopenia, epicardial adiposity, and increased intestinal permeability are common pathways for liver and myocardial fibrosis, in association with a proinflammatory environment [27,28,29]. They represent key points in AF physiopathology and future therapeutical targets.

Obesity, diabetes mellitus, and fatty liver disease are recognized as proinflammatory and profibrotic conditions that can be associated with heart fatty infiltration and localized conduction blocks [30,31], increasing the risk of AF in these special metabolic populations, along with the diastolic dysfunction associated with NASH.

Targher et al. demonstrated a high risk for AF in patients with diabetes mellitus and NASH compared to those without NASH. [32]. Another prospective study that followed, for 16.3 years, a large cohort of patients, reported that NASH was independently associated with atrial fibrillation (OR = 1.88, 95% CI 1.03–3.45) [20]. Moreover, recently it was demonstrated that increased liver fibrosis expressed by a high liver stiffness index evaluated by transient elastography, is associated with an increased risk of atrial fibrillation, advanced liver fibrosis being an independent risk factor for AF [33,34]. Recently, it was demonstrated that alterations in the oxidant/antioxidant balance have an impact on metabolism and cause cellular lipotoxicity, lipid peroxidation, chronic endoplasmic reticulum stress, and mitochondrial dysfunction. All these mechanisms are closely associated with chronic impairment of lipid metabolism, oxidative stress, and hepatic stelate cell activation [35]

Alcohol consumption is associated with an increased risk of AF development. Heavy drinkers (>21 drinks/week) have a risk ratio of 1.39 to developing AF [36]. These patients also associate with an increased risk of alcoholic liver disease, with rapid fibrosis progression to liver cirrhosis. Chronic alcohol consumption is also associated with an increased risk of alcoholic dilatative cardiomiopathy [37], which could be prone to AF. It has also been demonstrated that an increase in 10 g of alcohol consumption per day was associated with an increase of 0.16 mm in the left atrium diameter [38].

## 3. Particularities of the Anticoagulant Treatment in Liver Cirrhosis

Oral anticoagulation represents an important component of the integrative management of patients with AF, along with heart rate control, preventing the evolution of cardiac failure, and antiarrhythmic treatment [8]. According to the European Society of Cardiology guideline, AC should be considered for men with a CHA2DS2-VASc score of 2 and women with a score of 3, balancing the expected stroke reduction, bleeding risk, and patient preference [5]. The ideal anticoagulant treatment for this special population has not yet been recognized. There are a lot of particularities of these patients that could negatively influence the response to AC in decompensated LC: a large volume of distribution, low levels of proteins, impaired renal function, and sarcopenia [12,18].

The coagulation status in liver cirrhosis is the consequence of the balance between the decreased production of both procoagulant and anticoagulant factors, and it is very fragile especially in patients with decompensated LC, making the decision of starting an anticoagulant treatment very difficult [36]. Liver cirrhosis is associated with complex coagulation defects that involve primary and secondary coagulation processes and fibrinolysis [14,39].

Thrombocytopenia is a consequence of spleen sequestration and low levels of thrombopoetin [40], although the function of plateletes is not impaired because LC is characterized by an increased von Willebrand (vW) factor and factor VIII [41]. There is also evidence that hypercoagulability is a relatively frequent manifestation of protein C, protein S, and antithrombin III secondary deficiency in LC, as liver cirrhosis is characterized by decreases in both procoagulant factors and anticoagulant factors [39,42] (Figure 2). LC is also characterized by a fragile balance between pro antifybrinolitc agents, especially in decompensated cirrhotic patients complicated with infections or acute kidney injury [43].

There is evidence that, in cirrhotic patients, the markers of endothelial dysfunction, such as P-selectin or isoprostanes, are increased, suggesting that activated endothelial cells may favor thrombotic events in patients with decompensated LC [39]. In addition, patients with decompensated LC are more prone to bleeding secondary to endothelial dysfunction, production of heparin-like agents by bacterial overgrowth, acute kidney injury, or portal hypertension-related hemodynamic changes [44,45]. This balance between procoagulant and anticoagulant factors is a dynamic process, hence, the coagulation status of patients with LC at any given time may vary and could be very unpredictable, especially in the decompensated stage of the disease, and the best anticoagulant regimen is difficult to be chosen in patients with decompensated LC [42]. The common options for preventing thromboembolic complications in patients with AF and decompensated LC are vitamin K antagonists and direct oral anticoagulants.

### 3.1. Vitamin K Antagonists

VKAs inhibit vitamin K-dependent synthesis of clotting factors II, VII, IX, and X, and decrease the production of protein C and S inhibitors.

There are some disadvantages to using VKAs for thromboembolism prophylaxis complications in cirrhotic patients with AF because the coagulopathy secondary to liver disease frequently results in an elevated international normalized ratio (INR); thus, utilizing the INR to guide dosing of VKAs is particularly difficult in patients with LC [46,47]. In addition, the narrow therapeutic index (INR 1.8–2.20) and the significant drug–drug interactions represent important challenges.

A recent meta-analysis demonstrated that VKAs prophylaxis in patients with LC and AF was associated with an increased risk of bleeding complications, including portal hypertension bleeding (esophageal or gastric variceal bleeding) with almost the same efficacy in stroke prevention [18].

Considering the difficulties in treatment monitoring, labile INR in decompensated stages of liver cirrhosis, influence of the MELD score, and drug–drug interactions, interest has moved to DOACs [48].

### 3.2. Direct Oral Anticoagulants

DOACs directly and specifically target thrombin (dabigatran) or factor Xa (rivaroxaban, apixaban, or edoxaban) [49].

The pharmacokinetic properties of DOACs provide a reason to be cautious in using this treatment in patients with LC. Plasmatic clearance, plasma protein binding, biliary clearance, cytochrome P450 metabolism, and renal impairment can all be affected in liver cirrhosis [50]. Free drug fractions can also increase in patients with decompensated LC and low albumin levels [51]. Due to the decreased hepatic protein synthesis, protein-bound drugs such as rivaroxaban have low potency in patients with decompensated LC. Given these pharmacokinetic properties, all DOACs are contraindicated in Child–Pugh class C cirrhotic patients. Nevertheless, we have to take into account the risk of DOAC-induced liver injury (DILI) [52]. The highest risk of DILI is attributed to rivaroxaban, followed by dabigatran and apixaban [53].

There are pharmacokinetic differences among DOACs. Rivaroxaban and apixaban are mostly metabolized in the liver (67%), and they have half-lives between 5 h and 12 h after oral administration. Edoxaban is metabolized in the hepatocytes, and it has a half-life of between 15 and 20 h. Dabigatran is one of the DOACs with a very low rate of hepatic metabolism, and plasmatic proteins have no effect on its half-life, which is 12 to 14 h. It is, theoretically, the most perfect anticoagulant treatment for patients with LC [53]. Dabigatran also has an antagonist, idarucizumab (a monoclonal inhibitor antibody). Adexanet alfa is the antagonist to factor Xa inhibitors [52,53].

Over time, DOACs have proven their efficacy and safety in treatment and preventing thrombotic events [54]. DOACs have some advantages and some drawbacks in patients with LC. They have easy oral administration with no need for laboratory monitoring, and their anticoagulant activity is independent of antithrombin III level. Rivaroxaban and apixaban are more than 60% metabolized in the liver, with half-lives of 5–9 h and 12 h, respectively [55]. Edoxaban is 50% metabolized in the liver with a half-life of 10–15 h [55]. Dabigatran has minimal binding to plasma proteins and renal excretion with no hepatic metabolism with a half-life of 12–14 h [54].

Another advantage of DOACs is the development of an antidote, although the use of specific reversal agents in patients with AF and DOAC treatment should be restricted to severe bleeding complications. Idarucizumab is a humanized monoclonal antibody that is a specific antidote for dabigatran [56]. Andexanet alpha is a modified human recombinant factor Xa decoy protein that binds with high affinity with all direct factor Xa inhibitors and also LMWH and fondaparinux [57].

## 4. Anticoagulant Prophylaxis in Patients with Decompensated Liver Cirrhosis and Atrial Fibrillation

Anticoagulant treatment in patients with decompensated LC is still a matter of great debate because few data support the use of anticoagulant treatment in patients with decompensated LC.

### 4.1. Ischemic Complications Prophylaxis

The prevention of ischemic complications is the main aim of anticoagulant prophylaxis in patients with AF. This task could be very difficult in patients with decompensated LC, considering the limited data published until now (Table 1).

Several studies have demonstrated that the risk of ischemic stroke was lower in patients with LC receiving VKAs compared to no anticoagulant therapy [58,60]. In a prospective cohort, Pastori et al. [34] confirmed that patients with AF and advanced liver fibrosis, defined on prophylactic treatment with VKAs, have an increased risk of ischemic cardiovascular events, including ischemic stroke, compared with those treated with DOACs [18]. Until now, there have been no studies evaluating the role of anticoagulant treatment in stroke prevention in patients with AF and decompensated LC, although it has been demonstrated that cirrhotic men with CHA2DS2-VASc scores ≥2 and women with CHA2DS2-VASc scores ≥3 benefit more from the anticoagulant treatment in regard of thromboembolic events prevention [71].

### 4.2. Bleeding Risk Associated with the Anticoagulant Treatment

Bleeding complications in patients with decompensated LC related or not to portal hypertension are associated with an increased mortality rate and a high risk of developing acute-on-chronic liver failure. Therefore, efforts should be made to prevent this complication, including choosing the correct indication for anticoagulant treatment in patients with decompensated LC. Unfortunately, the bleeding risk score used in the general population is not applicable in patients with decompensated LC.

There have been several bleeding risk scores developed, mostly for patients treated with VKAs. These include the HAS-BLED score (hypertension, abnormal renal/liver function, stroke history, bleeding history or predisposition, labile INR, elderly, alcohol use) and the ORBIT score (Outcomes Registry for Better Informed Treatment of Atrial Fibrillation) [8,75]. The most used in clinical practice, including in patients with decompensated LC is the HAS-BLED score. Lee et al. [58] demonstrated a direct correlation between the HAS-BLED score and the Child–Pugh score, with the incidence of major bleeding being higher in patients with advanced liver cirrhosis. Recently, in a retrospective cohort, Efird et al. defined a risk stratification score, including serum creatinine and serum albumin levels [76]. This score could help clinicians to identify patients with decompensated LC and a high risk of bleeding complications following anticoagulant treatment, although it needs to be validated in larger prospective cohorts [76].

In patients with decompensated LC or a further decompensation stage (spontaneous bacterial peritonitis, hepato-renal syndrome, hepato-pulmonary syndrome) the benefits of DOACs have been demonstrated to diminish, possibly due to reduced drug metabolism and severely impaired hepatobiliary excretion [65] and VKA treatment was very difficult to be monitored. Sasso et al. [63] also confirmed that cirrhotic patients with esophageal varices treated with AC had a significantly higher risk (odds ratio 5.7, CI 1.8–17.7, *p* < 0.05) of bleeding than cirrhotic patients without esophageal varices. Moreover, Lee et al. demonstrated that the incidence of major bleedings was significantly higher in decompensated LC compared with those in a compensated stage of the liver disease (18% per year vs. 9.2% per year, *p* = 0.001) [58]. In addition, the multivariate Cox regression demonstrated that VKAs increased the risk of major bleeding events in patients with decompensated LC (adjusted HR 2.98, 95% CI 1.23–7.19, *p* = 0.002).

Five retrospective cohort studies compared the bleeding risk between VKAs and DOACs for thrombosis and stroke prevention in patients with AF with liver cirrhosis [17,59,61,64,65]. No significant difference in all-cause bleeding was observed between VKAs and DOACs during an almost 3-year follow-up period [17,59,61]. Goriacko et al. [61] evaluated patients with LC and AF receiving dabigatran. In accordance with the previous studies, they also found no difference regarding the bleeding risk in their cohort, including 55.4% decompensated LC patients.

Hum et al. [59] demonstrated that the risk of major bleeding complications was higher in patients receiving VKAs compared with DOACs, especially for intracerebral hemorrhage (*p* = 0.06), with no difference regarding gastrointestinal bleeding complications. Lee et al. [65] also demonstrated a statistically significant lower rate of major gastrointestinal bleeding in patients with DOACs compare to warfarin (1.9% vs. 3.6%, *p* = 0.003) and major bleeding (2.9% vs. 5.4%, *p* < 0.001). For patients with decompensated liver disease, the DOAC treatment decreased the risk of intracerebral hemorrhage compared to warfarin in the Asian population.

A meta-analysis including real-world studies evaluating AC in patients with LC and AF demonstrated that edoxaban treatment was associated with a higher risk of intracranial bleeding, although it was associated with the lowest risk of gastrointestinal bleeding [48]. Considering the balance between thrombotic event prevention and bleeding complications, the meta-analysis demonstrated that apixaban was the safest and most efficacy AC treatment for patients with decompensated LC and AF [48].

Two recent studies provided new data regarding the risk of bleeding in patients with decompensated LC receiving anticoagulant treatment. Oldham et al. demonstrated that patients with decompensated LC and AF developed more frequent bleeding complications compared to patients reported by the ARISTOTLE trial (67% vs. 29%) [73,77]. Concomitantly, Coons et al. demonstrated that Child–Pugh class C cirrhotic patients had a significantly higher bleeding risk during DOAC treatment for stroke prevention. This could sustain the need for a personalized bleeding risk scale in patients with LC and AF with high CHA2DS2-VASc scores.

Even if the AC treatment in patients with decompensated LC is still an off-label recommendation, it could be safe in Child–Pugh class B patients, although it is contraindicated in Child–Pugh class C patients. Apixaban and dabigatran are the first-line choices for stroke prevention in patients with decompensated LC and AF. For patients with Child–Pugh class C LC and a high CHA2DS2-VASc score, a low dose of apixaban or dabigatran could represent an option. Considering the frailty of cirrhotic patients and the important variability in their performance status, the AC treatment indication should be revised periodically and when the patients are included in Child–Pugh class C or a further decompensation stage, the AC treatment should be stopped. Screening for esophageal varices and primary prophylaxis should be started in every decompensated cirrhotic patient before DOAC treatment initiation.

Acute-on-chronic liver failure, sepsis, or acute kidney injury could also predispose the cirrhotic patient to bleeding complications [13,78]. In all these cases, the AC treatment should be stopped and restarted after the acute episode was treated and the patient could be included in the Child–Pugh class B or C.

Before invasive procedures, patients with liver cirrhosis should be treated with anticoagulant medication in accordance with the same recommendations as patients without LC. Imaging guidance is also advised for liver biopsy, central venous line insertion, and jugular puncture for tranjugular porto-systemic shunt placement [79].

High risk procedures of bleeding in patients with LC to be considered are: endoscopic polypectomy, endoscopic mucosal resection or endoscopic submucosal dissection, endoscopic dilatation of strictures in the upper or lower GI tract, endoscopic therapy of varices, percutaneous endoscopic gastrostomy, endoscopic ultrasound-guided sampling or with interventional therapy, esophageal or gastric radiofrequency ablation, and endoscopic retrograde cholangiopancreatography with sphincterotomy. For these patients, the DOACs should be stopped 3 days before the procedure, and for dabigatran 5 days before the procedure, if the creatinine clearance is <50 mL/min [80]. Heparin bridging therapy is recommended in patients with high risk of thrombotic events: AF and mitral stenosis, AF and previous stroke or transient ischemic attack, or patients with AF associating three or more of the following: diabetes mellitus, age >75 years, congestive cardiac failure, and hypertension. The DOACs should be restarted 2–3 days after the procedure [79,80]. If the patient undergoes a low risk bleeding procedure, the morning dose of DOACs should be omitted.

In patients with liver cirrhosis and bleeding complications related to portal hypertension, bleeding should be managed with portal hypertension-lowering measures and endoscopic therapy. In the case of failure to control hemorrhage, the decision to correct hemostasis should be considered on a case-by-case basis [79].

## 5. Unmet Needs

In most of the studies evaluating the safety and efficacy of VKAs or DOACs in patients with LC, the Child–Pugh class was not assessed; considering all that, the outcomes may be different in relation to the severity of liver cirrhosis.

Moreover, the CHA2DS2-VASc and HAS-BLED scores were not validated in patients with decompensated LC. Considering the coagulation particularities in patients with LC, the bleeding scores should include biomarkers such as plasma fibrin clot structure in order to personalize and increase the safety of the anticoagulant treatment in these patients, and liver cirrhosis severity should be included in the bleeding risk stratification systems.

The ideal anticoagulant for preventing thromboembolic complications in cirrhotic patients with AF has not yet been described. The clinicians must be especially mindful of the elevated risk of thromboembolism in the liver cirrhosis population with AF. Studies should also evaluate the impact of liver cirrhosis etiology on the risk of cerebrovascular complications in patients with AF, or the response to AC treatment in each patient subset. Large prospective trials are needed to evaluate the efficacy and safety of oral anticoagulant treatment in patients with decompensated liver cirrhosis and to define the optimal AC dose regimen in this vulnerable category.

No doubt, many advances have been made during the last decade regarding different aspects of the coagulation cascade and the role of AC in cirrhotic patients, although a lot of puzzle pieces are still missing from this big picture of AC treatment and AF, especially in decompensated liver disease.

The cirrhotic patients diagnosed with AF represent a high-risk group for both major bleeding and thromboembolism complications comparable to that of patients with significant renal impairment. In comparison to VKAs, DOACs significantly reduce the risk of severe bleeding, ICH, by ensuring appropriate protection from embolic events and not raising the risk of gastrointestinal bleeding. Moreover, a decrease in mortality in patients using DOACs may be shown, particularly in a specific group of patients, such as those with AF.

## 6. Conclusions

DOACs can be used for thromboembolism prophylaxis complications in patients with decompensated Child–Pugh class B LC and AF, despite the lack of randomized control trials confirming their safety and efficacy. Apixaban and dabigatran are the first-line choices for stroke prevention in patients with decompensated LC and AF. For patients with Child–Pugh class C LC and a high CHA2DS2-VASc score, a low dose of apixaban or dabigatran could represent an option. Periodic evaluation of the benefit/risk ratio should be planned in patients with LC receiving long-term anticoagulation.

## Figures and Tables

**Figure 1 diagnostics-13-01160-f001:**
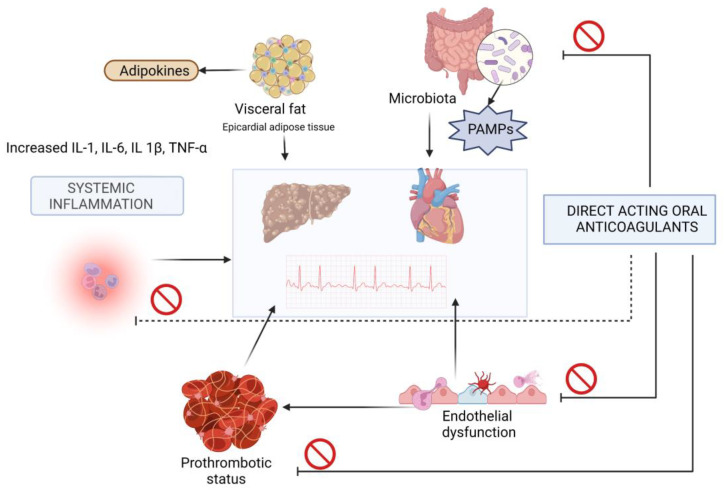
Physiopathology of atrial fibrillation in liver cirrhosis.

**Figure 2 diagnostics-13-01160-f002:**
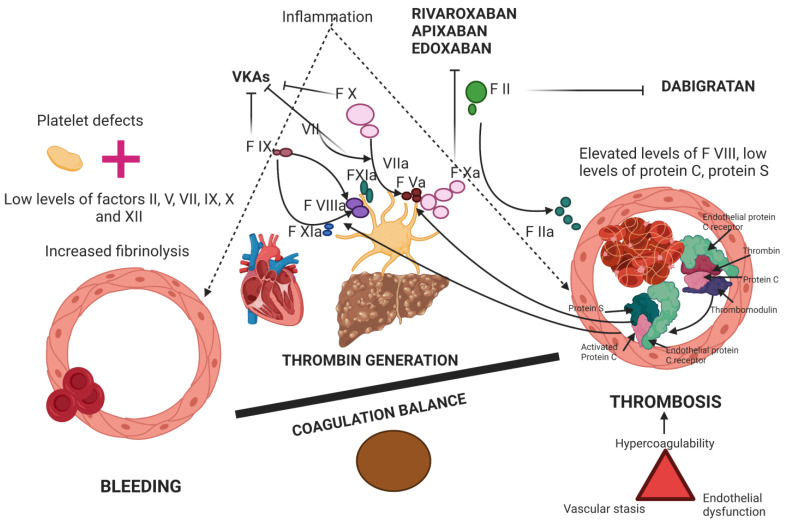
Coagulation balance in patients with liver cirrhosis.

**Table 1 diagnostics-13-01160-t001:** Anticoagulant treatment in patients with decompensated liver cirrhosis and atrial fibrillation.

Author, Year	Country	Type	Treatment	Patients	Age Mean/Median *	Mean Follow-Up	Child–Pugh Class B/C (%)	Outcomes
Lee SJ et al. [58], 2015	Korea	R, cohort	VKAs (INR 2–3)	173	62.1 years	2.3 years	62.4/37.6	Ischemic stroke 1.82% Major bleeding 9.61% ICH 1.82% GI bleeding non variceal 2.98% GI bleeding variceal 3.98 % AC treatment is not reducing clinical events in advanced LC
Hum J et al. [59], 2016	USA	R, cohort	DOACs (apixaban 5 mg BID, rivaroxaban 15 mg daily) VKAs (INR 2–3)	27 18	61.5 years 58.5 years	35 months	46.4/13.4	Major bleeding 4% vs. 28% ICH 0% vs. 17% GI bleeding 18% vs. 22%
Intagliata NM et al. [17], 2016	USA	R, cohort	DOACs (apixaban 5 mg BID/2.5 mg BID, rivaroxaban 20 mg/10 mg daily) VKAs (INR 2–3)	39 5	57 * years 60 * years	30.8 months	54/0	Major bleeding 4% vs. 10.5% ICH 5.0% vs. 5.26% GI bleeding 5.0% vs. 5.26% DOACs similar safety characteristics compared to VKAs
Kuo L et al. [60], 2017	Taiwan	R, cohort	VKAs (INR 2–3) No AC treatment	754	73.5 years	5 years	NA	VKAs use was associated with a lower risk of ischemic stroke The risk of ICH was similar between the two groups
Pastori D et al. [34], 2018	Italy	P, cohort	VKAs DOACs	77 52	74.4 years	3 years	NA	Major bleeding 14.2% vs. 1.92% ICH 3.89% vs. 1.92% GI bleeding 3.89% vs. 0% Higher bleeding events in patients treated with VKAs compared to DOACs
Goriacko P et al. [61], 2018	USA	R, cohort	VKA (INR 2–3) DOACs (apixaban 5 mg BID, rivaroxaban 20 mg daily, dabigatran 150 mg BID)	158 75	65 * years 66 * years	7 years	51.1/4.3	Major bleeding 15.8% vs. 13.3% No significant differences in bleeding complications rates between the two study groups
Qamar A et al. [62], 2019	USA	RCT	Edoxaban 30 mg/daily VKAs (INR 2–3)	-	68.4 years	2.8 years	NA	Ischemic stroke 1.07%/y Major bleeding 3.32%/y ICH 0.42%/y GI bleeding 1.26%/y Bleeding rate but not ischemic events rate was increased in patients with liver disease
Sasso R et al. [63], 2019	USA	R, case-control	VKA (INR 2–3) DOACs (apixaban 5 mg BID, rivaroxaban 20 mg daily, dabigatran 150 mg BID)	179	59 years	NA	49/7	Major bleeding 11.2% ICH 0.93% GI bleeding non variceal 11.2% GI bleeding variceal 4.37% Increased risk of variceal bleeding in patients with LC and AC treatment previously diagnosed with esophageal varices.
Lee SR et al. [64], 2019	Korea	R, cohort	VKA (INR 2–3) DOACs (apixaban 5 mg/2.5 mg BID, rivaroxaban 20 mg/10 mg daily, edoxaban 30 mg/15 mg daily, dabigatran 150/110 mg BID)	322 446	66.4 years 70.3 years	1.2 years	NA	DOACs higher safety profile and efficacy than VKAs in an Asian cohort
Lee H et al. [65], 2019	Taiwan	R, cohort	VKA (INR 2–3) DOACs (apixaban 2.5 mg BID, rivaroxaban 10 mg daily, dabigatran 110 mg BID)	990 1438	69.9 years 74.3 years	1.3 years	NA	DOACs the same efficacy as VKAs with a lower risk of major bleeding in an Asian cohort
Serper M et al. [66], 2020	USA	R, cohort	VKA DOACs	614 201	64.6 years 64.0 years	5 years	21.1/0.5	Ischemic stroke 2.3 vs. 1.3 per 100 person-year Major bleeding 5.9 vs. 3.6 per 100 person-year VKAs and DOACs are associated with decreased mortality. VKAs associate an increased risk of bleeding
Mort JF et al. [67], 2020	USA	R, cohort	DOACs (apixaban 5 mg BID, rivaroxaban 20 mg daily, dabigatran 150 mg BID)	44	-	427 days per patient	50.7/16.7	Major bleeding 8% ICH 0.7% GI bleeding non variceal 8% GI bleeding variceal 2.9% Patients with decompensated LC have significant bleeding rates
Davis KA et al. [68], 2020	USA	R, cohort	VKAs (INR 2–3) DOACs (apixaban 5 mg BID, rivaroxaban 20 mg daily, dabigatran 150 mg BID)	28 23	59 * years 63 * years	1 year	38.8/6	Ischemic stroke 0% vs. 1.8% Major bleeding 9.1% vs. 5.2% ICH- none in both groups GI bleeding 6.36% vs. 3.5% DOACs represent a safer alternative to VKAs in patients with mild and moderate LC
Jones K et al. [69], 2020	USA	R, cohort	VKAs (INR 2–3) DOACs (apixaban 2.5 mg/5 mg BID, rivaroxaban 20 mg/ 10 mg daily, dabigatran 110 mg BID)	29 33	70.3 years 71.9 years	4.5 years	34.2/2.5	Ischemic stroke 0% vs. 0% Major bleeding 5.4% vs. 2.4% No differences between VKAs and DOACs regarding the bleeding or the ischemic risk
Rusin G et al. [70], 2021	Poland	P, cohort	DOACs (apixaban 2.5 mg BID, rivaroxaban 10 mg/daily, dabigatran 110 mg BID)	42	65.4 years	4 years	45.2/0	Ischemic stroke 10.5% Major bleeding 10.5% Reduced doses of DOACs are safe in decompensated LC
Steensig G, et al. [71], 2022	Denmark	R, cohort	No AC OACs (apixaban 2.5 mg/5 mg BID, rivaroxaban 20 mg/ 10 mg daily, dabigatran 150/110 mg BID)	98 355	70 years 71 * years	3 years	NA	Risk of bleeding in cirrhotics on OACs 11.3% vs. 9.5% non cirrhotic on OACs
Coons EM, et al. [72], 2022	USA	R, cohort	VKAs (INR 2–3) DOACs (apixaban 5 mg BID, rivaroxaban 20 mg/ daily, dabigatran 150 mg BID)	41 44	63.6 years 67.2 years	6 years	54.5/18.2	Ischemic stroke 0% vs. 0% Major bleeding 14.6% vs. 9.1% ICH- none in both groups GI bleeding 14.6% vs. 20.5% Similar bleeding rates between VKAs and DOACs
Oldham M, et al. [73], 2022	USA	R, cohort	VKAs (INR 2–3) DOACs (apixaban 5 mg/2.5 mg BID, rivaroxaban 20 mg/15 daily, dabigatran 150 mg BID)	32 69	67 years 61 years	2 years	88.1/11.9	Ischemic stroke 4% vs. 0% Major bleeding 24% vs. 43% ICH- 0% vs. 14% GI bleeding 52% vs. 57%
Yoo SY, et al. [74], 2022	Korea	P, cohort	VKAs (INR 2–3) DOACs (apixaban 5 mg/2.5 mg BID, rivaroxaban 20 mg/15 daily, edoxaban 30 mg/15 mg daily, dabigatran 150 mg BID)	110 128	65.2 years 70.4 years	6 years	25.6/0	Ischemic stroke 4.2% Major bleeding 18.1% vs. 7.8% ICH- 0.9% vs. 0.78% GI bleeding 12.7% vs. 5.4%

Abbreviations: AC, anticoagulant; DOACs, direct oral anticoagulants; GI, gastrointestinal; ICH, intracerebral hemorrhage; LC, liver cirrhosis; OACs, direct oral anticoagulants; R, retrospective; VKAs, vitamin K antagonists. * median valueData are limited regarding the prevalence of these complications in patients with decompensated LC. In an Asian cohort, the authors demonstrated that patients with decompensated LC had the same frequency of ischemic stroke compared with patients withor without VKA treatment (2.56% vs. 2.5%, *p* = 0.98) [64]. Lee et al. [64] also demonstrated that the risk of ischemic stroke in patients with LC receiving DOACs was similar in a warfarin group (3.2 vs. 3.7%, *p* = 0.430), indicating that we need to be more cautious when considering an anticoagulant in patients with AF and decompensated LC.

## Data Availability

Not applicable.

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
