# Peer review of "Anticoagulation for Atrial Fibrillation in Patients with Decompensated Liver Cirrhosis: Bold and Brave?"

_diagnostics, 2023, doi:10.3390/diagnostics13061160_

Round 1

Reviewer 1 Report

Please find another presentation than table 1, a figure would be better

Author Response

We thank the Reviewer for considering our manuscript. Below are Reviewer 1’s specific comment and our response.

  1. Please find another presentation than table 1, a figure would be better

Response: Thank you for this suggestion. We added a figure to our manuscript. Please see figure 1, page 3.

Reviewer 2 Report

Girleanu and al. provided an intetersting work concerning anticoagulation in case of atrial fibrillation, in cirrhotic patients.

The paper is clear, references are accurate.

It would be interesting to include the management of AC in case of previous bleeding or before and after procedures, and the risk in case of AF.

Author Response

We thank the Reviewer for considering our manuscript. Below are Reviewer 2’s specific comment and our respons.

Girleanu and al. provided an intetersting work concerning anticoagulation in case of atrial fibrillation, in cirrhotic patients. The paper is clear, references are accurate

  1. It would be interesting to include the management of AC in case of previous bleeding or before and after procedures, and the risk in case of AF.

Response: Thank you for this suggestion. This has been added (please, see page 15).

Reviewer 3 Report

This is a comprehensive review regarding the safety and efficacy of anticoagulant treatment in decompensated cirrhosis patients with atrial fibrillation. It will be a good source for the audience to understand the application and anticoagulation treatment in LC patients.

Here are a few suggestions:

1. Line 114. "Alcohol consumption" should not be bold.

2. For VKAs and oral anticoagulants, what are the dosages used in each study? The authors could include these information in the review.

3. Line 179-180. More information may be needed for the working mechanism of DOACs. And what are the similarity or differences of rivaroxaban, apixaban, or edoxaban?

4. What are the age groups of patients in all the studies?

5. It will be helpful if the authors can add a few figures in this review, such as coagulation pathway or how to choose coagulant for LC patients with AF etc, so that the audience can better understand the concept.

6. Please read over carefully to check for grammatical errors and consistency. It is also very important to summarize all studies discussed in this review clearly so that confusion can be avoid.

Author Response

We thank the Reviewer for considering our manuscript. Below are Reviewer 3’s specific comments and our response to each one.

This is a comprehensive review regarding the safety and efficacy of anticoagulant treatment in decompensated cirrhosis patients with atrial fibrillation. It will be a good source for the audience to understand the application and anticoagulation treatment in LC patients. Here are a few suggestions:

  1. Line 114. "Alcohol consumption" should not be bold.

Response: Thank you for this suggestion. This has been corrected (please, see page 4)

  1. For VKAs and oral anticoagulants, what are the dosages used in each study? The authors could include these information in the review.

Response: We thank the Reviewer for these remarks. We added these data in Table I (please, see pages 7-13, Table I)

  1. Line 179-180. More information may be needed for the working mechanism of DOACs. And what are the similarity or differences of rivaroxaban, apixaban, or edoxaban?

Response: We fully agree with the Reviewer ‘s comments. This has been added (please, see page 7)

  1. What are the age groups of patients in all the studies?

Response- : Thank you for this suggestion. We added these data in Table I (please, see pages 7-13, Table I)

  1. It will be helpful if the authors can add a few figures in this review, such as coagulation pathway or how to choose coagulant for LC patients with AF etc, so that the audience can better understand the concept.

Response- We fully agree with the Reviewer‘s comments. This has been done (please see Figure 1 and Figure 2).

  1. Please read over carefully to check for grammatical errors and consistency. It is also very important to summarize all studies discussed in this review clearly so that confusion can be avoid.

ResponseWe agree with these comments. This has been done.

Reviewer 4 Report

The review article by Girleanu I et al (Diagnostics-2245632-2023) presents the real-life difficulties of anticoagulation in patients with liver cirrhosis and particularly in those in a decompensated state. The authors do a detailed and critical review of the existing literature and compare the efficacy and complications of the two main classes of anticoagulation, namely vitamin K inhibitors and direct oral anticoagulants. They present ways of approaching possible problems that may arise and propose ways to deal with them.

The review is well written and the use of English is adequate. In addition, the listed data can be extrapolated in other cases where the cirrhotic patient needs anticoagulation, such as in portal vein thrombosis or other similar cases.

General Comments:

1.    Typing and grammatical errors need correction.

2.    The brain is not the sole place of thromboembolism in AF. Emboli can reach any vascular system of any organs. Authors should refer to "ischemic stroke" only in case the reported reference includes only patients with stroke following AF. Otherwise, other terms, as "thromboembolic complications" etc, are more appropriate.

Major Comments:

1.    (Line 13): Please replace "stroke prevention" by "thromboembolic protection". Stroke presents one of the several possible sites of thromboembolism in atrial fibrillation.

2.    (Line 39): Please clarify whether references #3 & 4 report on populations of comparative age.

3.    (Line 105): References #30 & 31 appear to be irrelevant. Please check.

Minor Comments:

1.   (Line 107-108): The whole sentence appears to be incomplete. Something is missing.

2.   (Line 118): “...cardiomyopathy, which could predispose to AF”.

3.   (Line 137): “...fibrinolysis.”

Author Response

We thank the Reviewer for considering our manuscript. Below are Reviewer 4’s specific comments and our response to each one.

The review article by Girleanu I et al (Diagnostics-2245632-2023) presents the real-life difficulties of anticoagulation in patients with liver cirrhosis and particularly in those in a decompensated state. The authors do a detailed and critical review of the existing literature and compare the efficacy and complications of the two main classes of anticoagulation, namely vitamin K inhibitors and direct oral anticoagulants. They present ways of approaching possible problems that may arise and propose ways to deal with them.

The review is well written and the use of English is adequate. In addition, the listed data can be extrapolated in other cases where the cirrhotic patient needs anticoagulation, such as in portal vein thrombosis or other similar cases.

General Comments:

  1.  Typing and grammatical errors need correction

Response: We thank the Reviewer for these remarks. This has been done (please,  see the revised manuscript)

  1. The brain is not the sole place of thromboembolism in AF. Emboli can reach any vascular system of any organs. Authors should refer to "ischemic stroke" only in case the reported reference includes only patients with stroke following AF. Otherwise, other terms, as "thromboembolic complications" etc, are more appropriate.

Response: Thank you for this suggestion. We agree that the correct term is “thromboembolic complications”. This has been changed  (please, see pages 1, 2, 4 and 6).

Major Comments:

  1. (Line 13): Please replace "stroke prevention" by "thromboembolic protection". Stroke presents one of the several possible sites of thromboembolism in atrial fibrillation.

Response: This has been done (please, see page 1).

  1. (Line 39): Please clarify whether references #3 & 4 report on populations of comparative age.

Response- Both populations with atrial fibrillation, with or without liver cirrhosis, had similar age groups. This has been added in the manuscript (please, see page 1).

  1. (Line 105): References #30 & 31 appear to be irrelevant. Please check.

Response- The references were charged (please, see page 16).

Minor Comments:

  1. (Line 107-108): The whole sentence appears to be incomplete. Something is missing.

ResponseThe phrase was corrected (please, see page 4).

  1. (Line 118): “...cardiomyopathy, which could predispose to AF”.

Response: This was corrected “dilatative cardiomiopathy “(please see page 4).

  1. (Line 137): “...fibrinolysis.”

           Response- This was corrected (please, see page 5).

Reviewer 5 Report

Interesting manuscript (revision). The authors present a novel proposal and question regarding anticoagulation of atrial fibrillation in liver cirrhosis. The manuscript is well written, the structure has a

I have the following comments.

I. Major comments:

1. After the introduction, include a section on the criteria used to select the cited manuscripts (methodology).

2. Briefly describe the main causes of liver cirrhosis. For example, high alcohol intake or non-alcoholic fatty liver disease (obesity). Briefly describing the role of oxidative stress in the progression of liver damage. PMID: 34687092

3. What clinical projections do the findings have? This point should be included in the manuscript.

II. Minor comment:

1. Improve the wording of the study objective

2. I suggest including a figure that integrates the most relevant aspects.

Author Response

We thank the Reviewer for considering our manuscript. Below are Reviewer 4’s specific comments and our response to each one.

Interesting manuscript (revision). The authors present a novel proposal and question regarding anticoagulation of atrial fibrillation in liver cirrhosis. The manuscript is well written, the structure has a

I have the following comments.

  1. Major comments:
  2. After the introduction, include a section on the criteria used to select the cited manuscripts (methodology).

Response: Thank you for this suggestion. This has been added (please, see page 2).

  1. Briefly describe the main causes of liver cirrhosis. For example, high alcohol intake or non-alcoholic fatty liver disease (obesity). Briefly describing the role of oxidative stress in the progression of liver damage. PMID: 34687092

Response: We thank the Reviewer for these remarks. We completed our manuscript according to your valuable suggestions (please see pages 2, 3 and 4)

  1. What clinical projections do the findings have? This point should be included in the manuscript.

Response: We fully agree with the Reviewer ‘s comments. This has been emphasized in the manuscript (please, see page 13).

  1. Minor comments:

  1.  Improve the wording of the study objective

Response- this has been done (please, see page 2).

  1. I suggest including a figure that integrates the most relevant aspects.

Response- We fully agree with the Reviewer‘s comments . We added figure 1 (please, see page 3, figure 1).

Round 2

Reviewer 3 Report

The authors have addressed all the questions by the reviewer.

Reviewer 5 Report

Authors answered all my comments. Therefore, the manuscript can be accepted.